# ALLEVIATING THE EFFECT OF DATA IMBALANCE ON ADVERSARIAL TRAINING

## ABSTRACT

In this paper, we study adversarial training on datasets that obey the long-tailed distribution, which is practical but rarely explored in previous works. Compared with conventional adversarial training on balanced datasets, this process falls into the dilemma of generating uneven adversarial examples (AEs) and an unbalanced feature embedding space, causing the resulting model to exhibit low robustness and accuracy on tail data. To combat that, we propose a new adversarial training framework – Re-balancing Adversarial Training (REAT). This framework consists of two components: (1) a new training strategy inspired by the effective number to guide the model to generate more balanced and informative AEs; (2) a carefully constructed penalty function to force a satisfactory feature space. Evaluation results on different datasets and model structures prove that REAT can effectively enhance the model's robustness and preserve the model's clean accuracy.

## 1 INTRODUCTION

Adversarial attacks (Goodfellow et al., 2015; Madry et al., 2018) have become a serious threat to deep learning models, where the adversary crafts adversarial examples (AEs) by adding imperceptible perturbations to a clean input, to deceive the model into making wrong predictions. To mitigate adversarial attacks, a prominent way is adversarial training (Madry et al., 2018), which generates AEs and incorporates them into the training set to improve the model's *adversarial robustness* and capability of correcting AEs during inference.

However, existing efforts of adversarial training mainly focus on balanced datasets, while ignoring more realistic datasets obeying long-tailed distributions (Lin et al., 2017; Cao et al., 2019; Cui et al., 2019). Informally, training data subject to a long-tailed distribution has the property that the vast majority of the data belong to a minority of total classes (i.e., "head" classes), while the remaining data belong to other classes ("body" and "tail" classes) (Wang et al., 2017). This distinct nature yields new problems in adversarial training (see Section 2.2 for detailed explanations). First, it is difficult to produce uniform and balanced adversarial examples (AEs): AEs are mainly misclassified by the model into the head classes with overwhelming probabilities regardless of the labels of their corresponding clean samples. Second, the excessive dominance of head classes in the feature embedding space further compresses the feature space of tail classes. The mutual entanglement of the above two problems leads to the underfitting of tail classes in both adversarial robustness and clean accuracy, thus leading to unsatisfactory training performance.

To address these challenges, Wu et al. (2021) proposed RoBal, the first work (and the only work, to our best knowledge) towards adversarial training on datasets with long-tailed distributions. It is essentially a two-stage re-balancing adversarial training method. The first stage lies in the training process, where a new class-aware margin loss function is designed to make the model pay equal attention to data from head classes and tail classes. The second stage focuses on the inference process, where a pre-defined bias is added to the predicted logits vectors, thereby improving the prediction accuracy of samples from the tail classes. Moreover, RoBal constructs a new normalized cosine classification layer, to further improve models' accuracy and adversarial robustness.

While RoBal shows impressive results on a variety of datasets, it still has several limitations. First, the robustness of RoBal benefits mainly from gradient obfuscation (specifically, gradient vanishing) (Athalye et al., 2018) in the proposed new scale-invariant classification layer. This can be easily compromised by simply multiplying the logits by a constant, as the constant can increase the absolute value of gradients against gradient vanishing and correct the sign of gradients during AE generation (see Tables 6 and 7). Second, the designed class-aware margin loss ignores samples from body classes

and exclusively focuses on head and tail classes, which inevitably reduces the overall model accuracy. Detailed analysis and evaluations can be found in Sections 2.2 and 4.3.

To advance the practicality of adversarial training on long-tailed datasets, we design a new framework: Re-balancing adversarial training (REAT), which demonstrates higher clean accuracy and adversarial robustness compared to RoBal. Our insights come from the revisit of two key components in adversarial training: AE generation and feature embedding. Particularly, **for AE generation**, we force the generated AEs to be misclassified into each class as uniformly as possible, so that the information of the tail classes is sufficiently learned during adversarial training to improve the robustness. Our implementation is inspired by the effective number (Cui et al., 2019) in long-tailed recognition, which was proposed to increase the marginal benefits from data of tail classes. We generalize its definition to the AE generation process and propose a new Re-Balanced Loss (*RBL*) function. *RBL* dynamically adjusts the weights assigned to each class, which significantly improves the effectiveness of the original balanced loss. **For feature embedding**, it is challenging to balance the volume of each class's feature space, especially if the size of each class varies. To address this issue, we propose a Tail-sample-mining-based feature margin regularization (*TAIL*) approach. *TAIL* treats the samples from tail classes as hard samples and optimizes feature embedding distributions of tail classes and others. To better fit the unbalanced data distribution, we propose a *joint weight* to increase the contribution of tail features in the entire feature embedding space. Visualization results can be found in Appendix E.

We conduct comprehensive experiments on CIFAR-10-LT and CIFAR-100-LT datasets to demonstrate the superiority of REAT over existing methods. For instance, REAT achieves 67.33% clean accuracy and 32.08% robust accuracy under AutoAttack, which are 1.25% and 0.94% higher than RoBal. We also provide a theoretical analysis for the lower risk bound of a robust model trained on the unbalanced dataset in Appendix C

We want to emphasize that REAT is not a simple combination of previous works. Its superiority lies in the newly proposed modifications over these methods, to adapt to the long-tailed adversarial training scenario. We validate that simply integrating existing methods cannot achieve satisfactory results. For instance, without generalizing the definition of effective number from normal training to AE generation, the original term will result in much lower performance ("ENR" in Table 2).

## 2 Background and Motivation

### 2.1 Long-tailed Recognition

Data in the wild usually obey a long-tailed distribution (Lin et al., 2017; Cao et al., 2019; Cui et al., 2019), where most samples belong to a small part of classes. Models trained on long-tailed datasets usually give higher confidence to the samples from head classes, which harms the generalizability for the samples from the body or tail classes. It is challenging to solve such overconfidence issues under the long-tailed scenarios (Japkowicz & Stephen, 2002; He & Garcia, 2009; Buda et al., 2018). Several approaches have been proposed to achieve long-tailed recognition. For instance, (1) the re-sampling methods (Liu et al., 2009; Han et al., 2005; Ren et al., 2020) generate balanced data distributions by sampling data with different frequencies in the training set. (2) The cost-sensitive learning methods (Hong et al., 2021; Cui et al., 2019; Lin et al., 2017) modify the training loss with additional weights to balance the gradients from each class. (3) The training phase decoupling methods (Kang et al., 2020; 2021) first train a feature extractor on re-sampled balanced data, and then train a classifier on the original dataset. (4) The classifier designing methods (Kang et al., 2020; Wu et al., 2021) modify the classification layer with prior knowledge to better fit the unbalanced data. Morel details about related works are in Appendix A.

### 2.2 Long-tailed Adversarial Training

Adversarial training is a promising solution to enhance the model's robustness against AEs. Previous works mainly consider the balanced datasets. Discussions of these works can be found in Appendix A. When the training data become unbalanced, training a robust model becomes more challenging. As mentioned in Section 2.1, in long-tailed recognition, most data come from the head classes while the data of the tail classes are relatively scarce. This causes two consequences: unbalance in the output probability space and unbalance in the feature embedding space, which are detailed as follows.

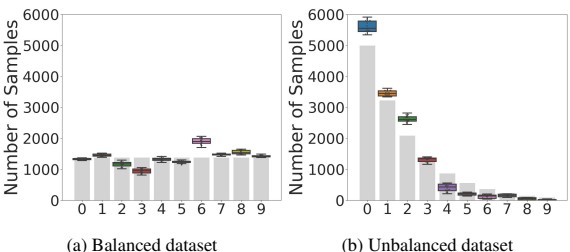 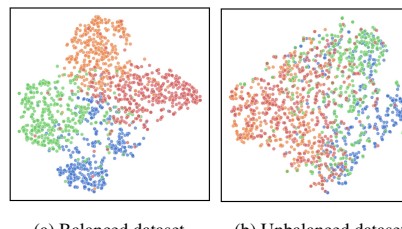

Figure 1: Prediction distributions of AEs. Clean label distributions are shown by gray bars.

Figure 2: Visualization of feature maps of AEs from models.

First, we need to generate AEs on-the-fly during adversarial training. The unbalanced output probability space caused by long-tailed datasets can lead to unbalanced AEs, which cause the produced model to show unbalanced robustness across different classes. Figure 1 shows such an example. We adopt PGD-based adversarial training to train a ResNet-18 model and measure the distribution of the model's predictions for the generated AEs during the training process. Figure 1a shows the case of a balanced training set (CIFAR-10). We observe that the predictions of the AEs are uniformly distributed among all the classes. In contrast, Figure 1b shows the case of an unbalanced training set (CIFAR-10-LT). Due to the long-tailed distribution, most AEs are labeled as head classes. This indicates that the final model has lower accuracy and robustness for tail classes, making it more vulnerable to adversarial attacks, e.g., AutoAttack (Croce & Hein, 2020).

Second, in an unbalanced training set, the head classes can dominate the feature embedding space of the model, which can reduce the area of tail features. As a result, the performance and generalizability of the model for tail classes will be decreased. In contrast, a model trained on balanced data will give an even feature space for each class. Figure 2 compares the feature maps of AEs in these two scenarios, where we train ResNet-18 models with PGD-based adversarial training on balanced and unbalanced CIFAR-10[1]. We observe the long-tailed scenario has larger differences between head and tail features compared to the balanced scenario.

A straightforward way is to directly adopt existing solutions introduced in Section 2.1 (e.g., (Lin et al., 2017; Cao et al., 2019; Cui et al., 2019; Ren et al., 2020)) for adversarial training, which can produce more balanced AE prediction distributions and feature embedding space. However, they can only partially address the overconfidence and underconfidence issues in model prediction, due to the lack of tail samples and AEs predicted as tail classes (see Section 4.3 and Appendix E). RoBal (Wu et al., 2021) is the first methodology dedicated to adversarial training with long-tailed datasets. It introduces a new loss function to promote the model to learn features from head classes and tail classes equally. It further replaces the traditional classification layer with a cosine classifier, where both weights and features are normalized and the outputs are multiplied by a temperature factor. In the inference phase, RoBal adjusts the output logits with a prior distribution, which is aligned with the label distribution. However, in our experiments, we find RoBal ignores the features from the body classes, which can harm the clean accuracy and robustness. Furthermore, RoBal can be easily defeated by a simple adaptive attack, which multiplies the output logits by a factor when generating AEs (see Section 4.3). This motivates us to explore a better solution for long-tailed adversarial training.

## 3 METHODOLOGY

We introduce REAT, a new framework for adversarial training on unbalanced datasets. REAT includes two innovations to address the two issues disclosed in Section 2.2. Specifically, to balance the AE distribution and encourage the model to learn more information from tail samples, we modify the objective function in the AE generation process with weights calculated based on the effective number (Cui et al., 2019). To balance the feature embedding space, we propose a regularization term to increase the area of features from tail classes.

**Preliminaries**. We first give formal definitions of a long-tailed dataset. Consider a dataset containing $C$ classes with $N_i$ samples in each class $i$. We assume the classes are sorted in the descending order based on the number of samples in each class, i.e., $N_i \geq N_{i+1}$. The unbalanced ratio is defined

---

[1]For better readability, we only show four classes (two head classes "airplane" (blue) and "automobile" (orange), and two tail classes "ship" (green) and "truck" (red).). The complete feature maps for 10 classes can be found in Appendix B.

as $\text{UR} = \frac{N_1}{N_C}$ (Cao et al., 2019). Following previous works (Wang et al., 2017; Cui et al., 2019), a long-tailed dataset can be divided into three parts: (1) $i$ is a head class (*HC*) if $1 \leq i \leq \lfloor \frac{C}{3} \rfloor$, where $\lfloor x \rfloor$ is a floor function; (2) $i$ is a tail class (*TC*) if $\lceil \frac{2C}{3} \rceil \leq i \leq C$, where $\lceil x \rceil$ is a ceiling function; (3) The rest are body classes (*BC*). Below, we describe the detailed mechanisms of REAT.

## 3.1 RE-BALANCING AEs

For adversarial training, it is desirable that the objective function could encourage AEs that are classified into rarely-seen classes while punishing AEs that are classified into abundant classes. To realize this in the long-tailed scenario, we borrow the idea of the effective number from (Cui et al., 2019) and generalize it to adversarial training. The effective number is mainly used to measure the data overlap of each class. For class $i$ containing $N_i$ data, its effective number is defined as $E_{N_i} = \frac{1-\beta^{N_i}}{1-\beta}$, where $\beta = \frac{\sum N_i - 1}{\sum N_i}$. Given the effective numbers $E_{N_i}$ and $E_{N_j}$, if $E_{N_i} > E_{N_j}$, the marginal benefit obtained from increasing the number of training samples in class $i$ is less than increasing the same number of training samples in class $j$ (Cui et al., 2019). This implies that we can adopt the effective number as a guide to balance the distribution of AEs generated during training.

**Motivation.** We explain the motivation of using the effective number in AE generation as follows. At a high level, *the generation of AEs can be viewed as a data sampling process*, i.e., AEs are essentially sampled from the neighbors of their corresponding clean samples. Therefore, we can calculate the effective number between AEs generated in two consecutive epochs, and use it as the basis to assign dynamic weights to each class in the loss function, inducing the model to produce as many less overlapped AEs as possible in consecutive epochs. This implicitly generates more AEs that are classified into tail classes and makes the model extract more marginal benefits from samples of tail classes, thus achieving our purpose.

**Technical design**. For simplicity, we assume that the predicted label distributions (i.e., labels assigned by the model $M$ for AEs) in two successive training epochs will stay stable and not change too much, which has been proven in (Zheng et al., 2020). Then, in epoch $k-1$, we count the number of AEs that are classified into each class, denoted as $\mathbf{n} = [n_1, n_2, \ldots, n_C]$.[2] As a result, generating AEs in epoch $k$ can be approximated as sampling new AEs after sampling $n_i$ data for each class $i$. Therefore, we can compute the effective number of class $i$ as $E_{n_i} = \frac{1-\beta_i^{n_i}}{1-\beta_i}$, where $\beta_i = \frac{N_i-1}{N_i}$. Note that our $\beta$ is *class-related* to assign finer convergence parameters for each class, which is different from the calculation in (Cui et al., 2019). We will experimentally prove that this adaptive effective number can better improve the model robustness in Section 4.1.

Based on the property that the effective number of each class is inversely proportional to the marginal benefit of the new samples of this class, we construct a new indicator variable weight $w_i$ for the marginal benefit, which is inversely proportional to $E_{n_i}$. This weight can be used to correct the loss in the AE generation process. Specifically, following the class-balanced softmax cross-entropy loss proposed in (Cui et al., 2019), we compute the weight $w_i$ for class $i$ as follows:

$$w_i = \frac{C}{E_{n_i} \sum_{j=1}^{C} \frac{1}{E_{n_j}}} \tag{1}$$

With the weight $w_i$ for each class $i$, we design a new Re-Balancing Loss (*RBL*) function as below:

$$RBL = -w_i * \log \frac{e^{z_i}}{\sum_j e^{z_j}} \tag{2}$$

where $\log \frac{e^{z_i}}{\sum_j e^{z_j}}$ is the original loss function adopted to generate AEs. Our goal is to maximize *RBL* to generate AEs for adversarial training.

**Analysis**. We analyze why *RBL* can help generate balanced AEs from unbalanced data samples. First, we show the effective number enjoys the *asymptotic properties*: (1) when $n_i \to 0$, we have $E_{n_i} \to 0$ and $w_i \to C$; (2) when $n_i \to \infty$, we have $E_{n_i} \to \frac{1}{1-\beta_i}$ and $w_i \to 0$, as there exists $E_{n_j} \to 0, i \neq j$. Based on the asymptotic properties, if there are many AEs assigned to the label of class $i$ in epoch $k-1$, then in epoch $k$, the increased effective number $E_{n_i}$ results in a smaller $w_i$. As

---

[2]For the first training epoch, we directly use the number of clean data $N_i$ in each class as the prior distribution.

a consequence, *RBL* will induce AEs generated in this round with minimized data overlap compared to AEs of the previous round, which implicitly generates more AEs that are classified into other classes. Our experiments in Section 4.3 indicate that combined with long-tailed recognition losses, *RBL* can better balance the AE generation process and increase the number of AEs predicted into tail classes by $5\times$. Figure 3 compares the distances of AEs between two consecutive training epochs without and with *RBL*. The models (ResNet-18) are trained on CIFAR-10-LT with the unbalanced ratio UR=50. A smaller distance indicates a larger overlap of the two AEs and less marginal benefit the model can obtain from the process. We observe that *RBL* is able to increase the distances of AEs from tail classes, and generate more informative AEs to enhance the model's robustness.

## 3.2 TAIL FEATURE ALIGNMENT

From Figure 2, we find the feature space of tail classes is smaller than that of head classes, making the model lean to classify the input into head classes. So, it is important to expand the feature space for tail classes to balance the feature representation. To achieve this goal, we first define a probabilistic feature embedding space as $\mathbf{f}^p = [\frac{e^{f_1}}{\sum_j e^{f_j}}, \frac{e^{f_2}}{\sum_j e^{f_j}}, \ldots, \frac{e^{f_K}}{\sum_j e^{f_j}}] = [f_1^p, f_2^p, \ldots, f_K^p]$, where $f_i$ is the $i$-th feature before the final classification layer, and $K$ is the feature dimension. The motivation of using a probabilistic feature embedding space is to overcome the scale changes in feature representations caused by the unbalanced data distribution (Wu et al., 2021). For each class $i$, we assume the probabilistic feature is sampled from a distribution $\mathcal{D}_i^f$. As a result, given any two classes $i \in TC$ and $j \in HC \cup BC$, our goal is to maximize the difference between $\mathcal{D}_i^f$ and $\mathcal{D}_j^f$, thereby rebalancing the distributions of different classes in the feature space and making them more divisible.

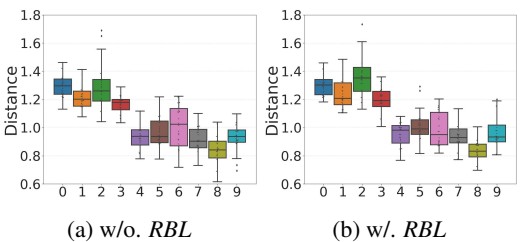

(a) w/o. *RBL*      (b) w/. *RBL*

Figure 3: Distributions of Euclidean distances of AEs generated from the same clean data in consecutive training epochs.

We design a Tail-sample-mining-based feature margin regularization (*TAIL*) approach to achieve this goal. Algorithm 1 describes its detailed mechanism. Specifically, let $\mathbf{F}^p = [\mathbf{f}_1^p, \mathbf{f}_2^p, \ldots, \mathbf{f}_B^p]$ denote all probabilistic features of a batch containing $B$ samples, and $\mathbf{y} = [y_1, y_2, \ldots, y_B]$ denote the labels of the corresponding feature representations. The class weights $\mathbf{\Omega} = [\omega_1, \omega_2, \ldots, \omega_C]$ are calculated based on the smoothed inverse class frequency (Mahajan et al., 2018; Mikolov et al., 2013), i.e., $\omega_i = \sqrt{\frac{\sum_j N_j}{N_i}}$, implying tail classes have larger class weights than head classes. The core component of *TAIL* is the computation of the regularization term $R$, which is updated for each $y_i \in TC$ using the following equation:

---

**Algorithm 1** *TAIL*

1: **Input:** probabilistic feature batch $\mathbf{F}^p$, label $\mathbf{y}$, class weights $\mathbf{\Omega}$, tail classes *TC*, batch size $B$
2: $R \leftarrow 0, S \leftarrow 0$
3: **for** $i = 1 \rightarrow B$ **do**
4:     **if** $y_i \in TC$ **then**
5:         $S = S + 1$
6:         Update $R$ following Equation 3
7:     **end if**
8: **end for**
9: **if** $S = 0$ **then**
10:     **return** 0
11: **else**
12:     **return** $\frac{R}{S}$
13: **end if**

---

$$R = R - \frac{1}{B} \sum_{j=1}^{B} (-1)^{\mathbb{1}(y_i = y_j)} (\omega_i + \omega_j) \sum_{k=1}^{K} f_{j,k}^p \log \frac{f_{j,k}^p}{f_{i,k}^p} \tag{3}$$

where $\mathbb{1}(y_i = y_j)$ is the indicator function (outputting 1 if $y_i = y_j$, and 0 otherwise), and the initial value of $R$ is 0. In Equation 3, we first compute the feature distribution differences using the Kullback–Leibler divergence (KLD): $\sum_{k=1}^{K} f_{j,k}^p \log \frac{f_{j,k}^p}{f_{i,k}^p}$, where $f_{i,k}^p$ is the value of the $k$-th dimension in the probabilistic feature $\mathbf{f}_i^p$ for the $i$-th sample. A larger KLD value means a larger difference between the distributions of the feature embeddings of the $i$-th and $j$-th samples. Hence, with the property of $R$, for each batch, we can maximize the distributional differences between $\mathcal{D}_i^f, i \in TC$ and $\mathcal{D}_j^f, j \neq i, j \in [C]$, and minimize the distributional gap for samples from the same tail class. To further enhance the influence of the regularization term among tail classes, we assign

a *joint weight* $(\omega_i + \omega_j)$ to the feature pair $(\mathbf{f}_i^p, \mathbf{f}_j^p)$. $\omega_i$ for tail samples is bigger than that for head samples. To increase the distinction between pairs of tail classes and non-tail classes and pairs of two tail classes, the joint weight further strengthens the effect of the regularization for pairs of two tail classes, thus improving the performance. Finally, we adopt the average distance inside the batch: $\frac{R}{S}$.

Note that our regularization term *TAIL* is general and can be used with any other long-tailed recognition loss function $L_{\mathrm{lt}}$ in the following form:

$$L = L_{\mathrm{lt}} + TAIL$$

To summarize, the training pipeline of our `REAT` uses *RBL* to generate adversarial examples, from which it trains the robust model with the loss function $L$.

## 4 EXPERIMENTS

**Datasets and Models**. We evaluate `REAT` on CIFAR-10-LT and CIFAR-100-LT, which are the mainstream datasets for evaluating long-tailed recognition tasks (Cui et al., 2019; Cao et al., 2019; Ren et al., 2020; Wu et al., 2021). To generate the unbalanced dataset, we follow the approach in (Cao et al., 2019) to set the unbalanced ratio (UR) as $\{10, 20, 50, 100\}$ for CIFAR-10-LT and $\{10, 20, 50\}$ for CIFAR-100-LT. We choose ResNet-18 (ResNet) (He et al., 2016) and WideResNet-28-10 (WRN) (Zagoruyko & Komodakis, 2016) as the target models.

**Baselines**. We consider two baselines. The first one is to *simply combine existing adversarial training methods with various long-tailed recognition losses*. Our experiments and analysis in Appendix B show that some adversarial training methods cannot converge well with the long-tailed recognition loss, such as TRADES (Zhang et al., 2019), AWP (Wu et al., 2021) and MART (Wang et al., 2020b). So we choose the most effective one: PGD-AT (Madry et al., 2018). The second baseline is RoBal (Wu et al., 2021).

**Implementation**. In our experiments, the number of training epochs is 80. The learning rate is 0.1 at the beginning and decayed in epochs 60 and 75 with a factor of 0.1. The weight decay is 0.0005. We adopt SGD to optimize the model parameters with a batch size of 128. We save the model with the highest robustness on the test set. For adversarial training, we adopt $l_\infty$-norm PGD (Madry et al., 2018), with a maximum perturbation size $\epsilon = 8/255$ for 10 iterations, and step length $\alpha = 2/255$ in each iteration. For each configuration, we report the mean and standard error under three repetitive experiments with different random seeds. Training with `REAT` is efficient and does not incur significant costs, as demonstrated in Appendix G.

**Attacks**. We mainly consider the $l_\infty$-norm attacks to evaluate the model's robustness. The results under the $l_2$-norm attacks can be found in Appendix D. We choose four representative attacks: PGD attack (Madry et al., 2018) with the cross-entropy loss under 20 and 100 steps (PGD-20 and PGD-100), PGD attack with the C&W loss (Carlini & Wagner, 2017) under 100 steps (CW-100), and AutoAttack (Croce & Hein, 2020) (AA).

### 4.1 ABLATION STUDIES

**Impact of Long-tailed Recognition Losses**. `REAT` is general and can be combined with different long-tailed recognition losses. We select four state-of-the-art losses and add each one with *TAIL* to evaluate `REAT`: focal loss (FL) (Lin et al., 2017), effective number loss (EN) (Cui et al., 2019), label-distribution-aware margin loss (LDAM) (Cao et al., 2019), and balanced softmax loss (BSL) (Ren et al., 2020). For comparisons, we also choose PDG-AT and replace the original cross-entropy loss with the above long-tailed recognition loss for model parameter optimization.

Table 1 shows the comparison results with ResNet-18 and CIFAR-10-LT (UR=50). We obtain two observations. (1) BSL loss can significantly outperform other long-tailed recognition losses for clean accuracy as well as adversarial robust accuracy against different attacks. So in the rest of our paper, *we mainly adopt this choice for evaluations*. (2) `REAT` achieves better adversarial robustness than PGD-AT for whatever loss function is adopted to train the model. Furthermore, the clean accuracy is improved in most cases when `REAT` is used. Therefore, we conclude that `REAT` has strong generalization and applicability to different recognition losses.

**Impact of AE Generation Re-balancing Losses**. We then compare the effectiveness of our *RBL* with various rebalancing methods adopted in the AE generation process. We replace the cross-entropy with four SOTA rebalancing strategies: (1) ReWeight (RW) (Huang et al., 2016; Wang et al., 2017);

| Losses | Method | Clean Accuracy | PGD-20 | PGD-100 | CW-100 | AA |
|--------|--------|----------------|--------|---------|--------|-----|
| FL | PGD-AT | 53.58(0.81) | 30.88(0.24) | 30.85(0.25) | 28.48(0.59) | 27.00(0.60) |
|    | REAT | **55.22**(1.42) | **31.14**(0.34) | **31.08**(0.33) | **28.71**(0.53) | **27.23**(0.56) |
| EN | PGD-AT | **55.26**(0.38) | 31.82(0.36) | 31.75(0.40) | 29.91(0.27) | 28.26(0.22) |
|    | REAT | 55.25(0.87) | **32.20**(0.25) | **32.14**(0.23) | **30.12**(0.33) | **28.69**(0.48) |
| LDAM | PGD-AT | 52.74(0.71) | 31.31(0.25) | 31.24(0.23) | 29.41(0.42) | 28.03(0.37) |
|      | REAT | **53.47**(1.04) | **31.52**(0.27) | **31.52**(0.27) | **29.63**(0.25) | **28.20**(0.21) |
| BSL | PGD-AT | 66.99(0.17) | 35.23(0.45) | 35.01(0.43) | 33.17(0.37) | 31.15(0.49) |
|     | REAT | **67.33**(0.45) | **36.20**(0.06) | **36.02**(0.09) | **33.98**(0.23) | **32.08**(0.12) |

Table 1: Results on CIFAR-10-LT (UR=50) with different long-tailed recognition losses. For this and the following tables, standard errors are shown inside (). Best results are in bold.

| Rebalancing | Method | Clean Accuracy | PGD-20 | PGD-100 | CW-100 | AA |
|-------------|--------|----------------|--------|---------|--------|-----|
| – |  | 66.99(0.17) | 35.23(0.45) | 35.01(0.43) | 33.17(0.37) | 31.15(0.49) |
| RW |  | 66.82(0.40) | 35.80(0.05) | 35.65(0.08) | 33.29(0.32) | 31.40(0.31) |
| RWS | PGD-AT | 67.28(0.63) | 35.83(0.35) | 35.70(0.37) | 33.38(0.50) | 31.50(0.67) |
| ENR |  | 66.53(0.91) | 35.26(0.14) | 35.08(0.12) | 32.95(0.27) | 31.05(0.13) |
| BRW |  | **67.98**(0.09) | 34.65(0.30) | 34.47(0.32) | 33.55(0.33) | 31.36(0.40) |
| *RBL* | PGD-AT | 67.46(0.65) | 35.59(0.18) | 35.48(0.19) | 33.51(0.39) | 31.68(0.33) |
| *RBL* | *TAIL* | 67.33(0.45) | **36.20**(0.06) | **36.02**(0.09) | **33.98**(0.23) | **32.08**(0.12) |

Table 2: Comparisons between different AE generation re-balancing strategies. *BSL loss is adopted.*

| UR | Method | Clean Accuracy | PGD-20 | PGD-100 | CW-100 | AA |
|----|--------|----------------|--------|---------|--------|-----|
| 10 | PGD-AT | **75.27**(0.32) | 42.66(0.20) | 42.36(0.20) | 41.18(0.21) | 38.81(0.10) |
|    | REAT | 75.20(0.03) | **42.97**(0.17) | **42.76**(0.19) | **41.52**(0.22) | **39.25**(0.21) |
| 20 | PGD-AT | 72.31(0.24) | 39.79(0.31) | 39.61(0.30) | 38.42(0.06) | 36.18(0.03) |
|    | REAT | **72.73**(0.50) | **40.57**(0.15) | **40.41**(0.12) | **38.55**(0.29) | **36.53**(0.21) |
| 50 | PGD-AT | 66.99(0.17) | 35.23(0.45) | 35.01(0.43) | 33.17(0.37) | 31.15(0.49) |
|    | REAT | **67.33**(0.45) | **36.20**(0.06) | **36.02**(0.09) | **33.98**(0.23) | **32.08**(0.12) |
| 100 | PGD-AT | 62.70(0.52) | **32.91**(0.17) | **32.73**(0.19) | 30.45(0.15) | 28.60(0.21) |
|     | REAT | **63.92**(0.68) | 32.84(0.07) | 32.69(0.15) | **30.73**(0.38) | **28.90**(0.33) |

Table 3: Results on CIFAR-10-LT with different values of UR. *BSL loss is adopted.*

(2) ReWeight Smooth (RWS) (Mahajan et al., 2018; Mikolov et al., 2013); (3) Effective Number Reweight (ENR) (Cui et al., 2019); (4) Balanced Softmax ReWeight (BRW) (Ren et al., 2020).

Table 2 shows the comparison results on CIFAR-10-LT (UR=50) with the ResNet-18 model structure. We observe that the considered strategies can indeed increase the clean accuracy and robustness of the final models by re-balancing the generated AEs. Particularly, our *RBL* outperforms other approaches, giving better robustness under different attacks. Furthermore, our dynamic effective number achieves better results than the original effective number implementation (ENR), in which the model adopts labels of the clean data to balance the AE generation. This is because our adaptive effective number allows the samples to equally learn features of both head and tail classes, and makes the model obtain more marginal benefit from the AEs. By combining *RBL* and *TAIL*, REAT achieves the best results under various attacks, which proves the effectiveness of the feature distribution alignment strategy.

## 4.2 EVALUATION UNDER VARIOUS SETTINGS

**Varying the Unbalanced Ratio**. We first investigate the impact of the unbalanced ratio on training performance. Table 3 shows the comparison results between PGD-AT and REAT on CIFAR-10-LT and ResNet-18. We have the following observations. (1) For both methods, increasing UR can reduce the model's clean accuracy and robustness. (2) REAT outperforms PGD-AT under different values of UR and attacks, due to the re-balanced AE generation and feature embedding space.

**Varying Datasets and Model Architectures**. REAT can well generalize to different datasets and models. Table 4 compares PGD-AT and REAT on CIFAR-100-LT with ResNet-18. Table 5 compares two approaches on CIFAR-10-LT and CIFAR-100-LT with two model architectures. Similar to the above results, REAT can bring additional performance improvement under various unbalance degrees and attacks for different configurations. More results and analysis can be found in Appendix C.

| UR | Method | Clean Accuracy | PGD-20 | PGD-100 | CW-100 | AA |
|---|---|---|---|---|---|---|
| 10 | PGD-AT | **45.96**(0.49) | 18.85(0.19) | 18.73(0.17) | 17.70(0.13) | 16.21(0.13) |
| | REAT | 45.94(0.15) | **19.26**(0.18) | **19.16**(0.18) | **17.99**(0.09) | **16.58**(0.06) |
| 20 | PGD-AT | **42.45**(0.53) | 16.36(0.13) | 16.24(0.14) | 15.47(0.17) | 14.17(0.09) |
| | REAT | 41.98(0.21) | **16.84**(0.10) | **16.72**(0.12) | **15.77**(0.23) | **14.45**(0.20) |
| 50 | PGD-AT | **37.70**(0.12) | 13.95(0.07) | 13.86(0.05) | 13.17(0.11) | 12.10(0.02) |
| | REAT | 37.43(0.37) | **14.25**(0.22) | **14.18**(0.26) | **13.38**(0.15) | **12.32**(0.17) |

Table 4: Results on CIFAR-100-LT with different values of UR. *BSL loss is adopted.*

| Method | | | Clean Accuracy | PGD-20 | PGD-100 | CW-100 | AA |
|---|---|---|---|---|---|---|---|
| CIFAR-10-LT | ResNet | PGD-AT | 66.99(0.17) | 35.23(0.45) | 35.01(0.43) | 33.17(0.37) | 31.15(0.49) |
| | | REAT | **67.33**(0.45) | **36.20**(0.06) | **36.02**(0.09) | **33.98**(0.23) | **32.08**(0.12) |
| | WRN | PGD-AT | 72.38(0.30) | 35.93(0.10) | 35.64(0.04) | 34.93(0.14) | 32.84(0.19) |
| | | REAT | **72.58**(0.31) | **36.53**(0.31) | **36.35**(0.32) | **35.30**(0.37) | **33.37**(0.37) |
| CIFAR-100-LT | ResNet | PGD-AT | **45.96**(0.49) | 18.85(0.19) | 18.73(0.17) | 17.70(0.13) | 16.21(0.13) |
| | | REAT | 45.94(0.15) | **19.26**(0.18) | **19.16**(0.18) | **17.99**(0.09) | **16.58**(0.06) |
| | WRN | PGD-AT | **50.07**(0.25) | 20.79(0.39) | 20.69(0.38) | 20.17(0.27) | 18.32(0.28) |
| | | REAT | 49.99(0.18) | **20.85**(0.16) | **20.71**(0.20) | **20.18**(0.09) | **18.35**(0.17) |

Table 5: Results on two datasets (UR=50) with different model structures. BSL loss is adopted.

## 4.3 COMPARISONS WITH ROBAL

To the best of our knowledge, RoBal (Wu et al., 2021) is the only work specifically focusing on adversarial training on unbalanced datasets. As analyzed in Section 2.2, there are several limitations in RoBal. Besides, we find that the scale-invariant classification layer in RoBal can cause gradient vanishing when generating AEs with the cross-entropy loss. It is because the normalized weights of the classification layer and the normalized features greatly reduce the scale of the gradients, making it fail to generate powerful AEs. *We propose a simple adaptive attack to break the gradient vanishing and invalidate RoBal*: the adversary can multiply the output logits by a factor (10 in all cases) when generating AEs, and then use these AEs to attack RoBal. This can significantly decrease the adversarial robustness of the trained model.

We perform experiments to compare RoBal and REAT from different perspectives, as shown in Tables 6 and 7. We adopt the CIFAR-10-LT and CIFAR-100-LT with ResNet-18 settings, respectively. More results with different configurations can be found in Appendix E. First, for PGD-based attacks, we show that the model robustness partially originates from the gradient vanishing, and our adaptive attack can successfully break this effort. CW attack and AA can easily break the gradient obfuscation in the classification layer, due to the different loss functions in the AE generation process. Second, comparing the results of RoBal and REAT under different values of UR, REAT can achieve higher clean accuracy and robustness, especially with larger UR. This indicates REAT is a better training strategy for highly unbalanced datasets.

Furthermore, Figure 4a illustrates the robust accuracy of ResNet-18 on CIFAR-10-LT (UR=50) under different numbers of PGD attack steps. It proves that our REAT outperforms RoBal under all attack budgets. Figure 4b plots the accuracy of ResNet-18 on CIFAR-10-LT (UR=50) under the PGD-20 attack for each class. Our REAT can achieve higher clean accuracy and robust accuracy on "body" classes, which will be explained below. More results can be found in Appendix E.

**Interpretation**. We perform an in-depth analysis of the comparisons between RoBal and REAT. We show the distributions of the predicted labels of AEs during adversarial training for these two approaches in Figure 5. We choose the configurations of CIFAR-10-LT (UR=50) and ResNet-18. Results under other configurations can be found in Appendix E. For RoBal, we observe that there are fewer AEs classified into body classes and more AEs classified into tail classes, indicating that RoBal makes the model pay more attention to head and tail classes while overlooking the body classes. In contrast, REAT treats the body and tail classes more equally, and this is one reason to achieve better performance on the "body" classes. Furthermore, we plot the feature embedding space with the t-SNE tool in Appendix E for feature-level comparison.

**Summary**. First, from the results, our solution brings more benefit than RoBal, as shown in Table 6. RoBal can sometimes hurt the robustness and clean accuracy. Second, we prove the robustness from RoBal partially depends on the gradient obfuscation and can be defeated by an adaptive attack.

| UR | Method | Clean Accuracy | PGD-20 | PGD-100 | CW-100 | AA |
|---|---|---|---|---|---|---|
| 10 | RoBal | **75.33**(0.39) | 45.98(0.39)
Adaptive: 41.25 | 45.97(0.39)
Adaptive: 41.13 | 41.02(0.02) | **39.30**(0.10) |
| | REAT | 75.20(0.03) | **42.97**(0.17) | **42.76**(0.19) | **41.52**(0.22) | 39.25(0.21) |
| 20 | RoBal | 71.92(0.62) | 43.23(0.25)
Adaptive: 38.45 | 43.19(0.22)
Adaptive: 38.20 | 38.23(0.07) | 36.17(0.28) |
| | REAT | **72.73**(0.50) | **40.57**(0.15) | **40.41**(0.12) | **38.55**(0.29) | **36.53**(0.21) |
| 50 | RoBal | 66.08(0.69) | 38.46(0.18)
Adaptive: 33.54 | 38.44(0.11)
Adaptive: 33.20 | 33.90(1.72) | 31.14(0.44) |
| | REAT | **67.33**(0.45) | **36.20**(0.06) | **36.02**(0.09) | **33.98**(0.23) | **32.08**(0.12) |
| 100 | RoBal | 60.11(0.62) | 36.08(0.18)
Adaptive: 30.55 | 36.05(0.20)
Adaptive: 30.36 | 30.57(0.77) | 28.64(0.28) |
| | REAT | **63.92**(0.68) | **32.84**(0.07) | **32.69**(0.15) | **30.73**(0.38) | **28.90**(0.33) |

Table 6: Results on CIFAR-10-LT with different values of UR. Red numbers represent the results under our adaptive attack. BSL loss is adopted for REAT.

| UR | Method | Clean Accuracy | PGD-20 | PGD-100 | CW-100 | AA |
|---|---|---|---|---|---|---|
| 10 | RoBal | 43.47(0.31) | 20.55(0.20)
Adaptive: 18.49 | 20.49(0.20)
Adaptive: 18.31 | **18.12**(0.23) | **16.86**(0.10) |
| | REAT | **45.94**(0.15) | **19.26**(0.18) | **19.16**(0.18) | 17.99(0.09) | 16.58(0.06) |
| 20 | RoBal | 39.58(0.40) | 17.73(0.11)
Adaptive: 16.00 | 17.71(0.08)
Adaptive: 15.93 | 15.58(0.11) | **14.55**(0.10) |
| | REAT | **41.98**(0.21) | **16.84**(0.10) | **16.72**(0.12) | **15.77**(0.23) | 14.45(0.20) |
| 50 | RoBal | 34.24(0.54) | 14.77(0.09)
Adaptive: 13.22 | 14.77(0.10)
Adaptive: 13.18 | 12.77(0.10) | 12.02(0.08) |
| | REAT | **37.43**(0.37) | **14.25**(0.22) | **14.18**(0.26) | **13.38**(0.15) | **12.32**(0.17) |

Table 7: Results on CIFAR-100-LT with different URs. Red numbers represent the results under our adaptive attack. BSL loss is adopted for REAT.

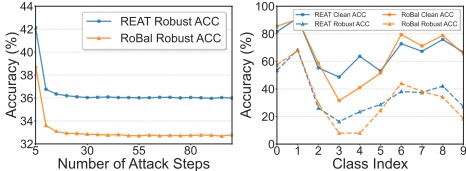

(a) Acc. of attack steps.  (b) Acc. of each class.

Figure 4: More comparisons results between REAT and RoBal for clean accuracy and adversarial robustness.

(a) RoBal            (b) REAT

Figure 5: Distributions of model predictions for AEs during training. Clean label distributions are shown by gray bars.

Third, through the results of PGD-AT with BSL loss, RoBal, and REAT, we find that compared with improving the robustness, it is easier to enhance the clean accuracy, which is consistent with the conclusion from (Wu et al., 2021). To explore the reason behind this phenomenon, we analyze the difficulty and challenges of adversarial training on long-tailed datasets in Appendix C. All in all, improving robustness requires more data, and the number of data in the tail classes is not enough to train a model with high robustness, which is a big challenge in adversarial training. How to further improve it is our future work.

## 5 CONCLUSION

In this paper, we propose REAT, a new long-tailed adversarial training framework to improve the training performance on unbalanced datasets. We present two novel components, *RBL* for promoting the model to generate balanced AEs, and a regularization term *TAIL* for forcing the model to assign larger feature spaces for tail classes. With these techniques, REAT helps models achieve state-of-the-art results and outperforms existing solutions on different datasets and model structures. There still exists a robustness gap between the ideal result obtained in the balanced setting and our approach. In the future, we aim to keep reducing this gap with more advanced solutions, e.g., new robust network structures or training loss functions.

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

## A    RELATED WORKS

### A.1    LONG-TAILED RECOGNITION

Long-tailed learning means training a machine learning model on a dataset that follows a long-tailed distribution. It has been applied to various scenarios including classification tasks (Ren et al., 2020), object detection tasks (Lin et al., 2017) and segmentation tasks (Wang et al., 2020a). To alleviate the uneven distribution of data in the dataset, i.e., the majority of the data belong to the head classes, while the data belonging to the tail classes are insufficient, many methods have been proposed, which can be roughly divided into four categories:re-sampling, cost-sensitive learning, training phase decoupling and classifier designing.

The re-sampling methods can be divided into four classes, i.e., random under-sampling head classes (Liu et al., 2009), random over-sampling tail classes (Han et al., 2005), class-balanced

re-sampling (Ren et al., 2020) and scheme-oriented sampling (Huang et al., 2016). These methods solve the unbalance problem by using sampling strategies to generate desired balanced distributions.

The cost-sensitive learning methods have two types of applications, i.e., class-level re-weighting (Hong et al., 2021; Cui et al., 2019; Lin et al., 2017) and class-level re-margining (Cao et al., 2019; Khan et al., 2019; Wu et al., 2021). It assigns different weights to each class or adjust the minimal margin between the features and the classifier to balance the learning difficulties, achieving better performance under unbalanced data distributions.

The training phase decoupling is used to improve both the feature extractor and classifier. Kang et al. (2020) find that training the feature extractor with instance-balanced re-sampling strategy and re-adjusting the classifier can significantly improve the accuracy in long-tailed recognition. Kang et al. (2021) further observe that a balanced feature space benefits the long-tailed recognition.

The classifier designing aims to address the biases that the weight norms for head classes are larger than them of tail classes (Yin et al., 2019) in the traditional layers under long-tailed datasets. Kang et al. (2020) propose a normalized classification layer to re-balance the weight norms for all classes. Wu et al. (2021) also adopt a normalized classifier to defend against adversarial attacks. Wu et al. (2020b) propose a hierarchical classifier mapping the images into a class taxonomic tree structure. Tang et al. (2020) propose a classifier with causal inference to better stabilize the gradients. Note that modifying the classifier can indeed improve the performance of models on unbalanced data. However, we argue that it may introduce gradient obfuscation resulting in adaptive adversarial attacks. For more details, please refer to Section 4.

## A.2 ADVERSARIAL TRAINING

Adversarial training (Madry et al., 2018; Zhang et al., 2019; Wang et al., 2020b; Rice et al., 2020) is widely studied to defend against adversarial attacks. Its basic idea is to generate on-the-fly AEs to augment the training set. It can be formulated as the following min-max problem (Madry et al., 2018):

$$\min_{\theta} \max_{x^*} \ell(x^*, y; \theta)$$

where $x^*$ is the training sample generated from a clean one $x$ to maximize the loss function $\ell(\cdot)$, $y$ is the ground-truth label, $\theta$ is the model parameters. The first phase (maximization optimization) is to generate samples maximizing the loss function. The second stage (minimization optimization) is to optimize the model parameter $\theta$ to minimize the loss function under samples generated in phase one.

In previous works, there are three main research topics in adversarial training, i.e., improving the model robustness (Madry et al., 2018; Wang et al., 2020b), reducing the gap between clean accuracy and robustness (Zhang et al., 2019; Wu et al., 2020a) and addressing overfitting challenges (Rice et al., 2020; Huang et al., 2020).

In this paper, we focus on adversarial training on datasets with long-tailed distributions. To our best knowledge, Wu et al. (2021) present the first work dedicated to improving the accuracy as well as robustness to tail class during adversarial training. They design a new loss function and cosine classifier to achieve this. However, we experimentally demonstrate the unsatisfactory security and performance of this work in Section 4.3, which motivates us to design more secure and satisfactory adversarial training methods tailored to datasets that obey long-tailed distributions.

## B ADVERSARIAL TRAINING ON UNBALANCED DATASET

To explore the effectiveness of adversarial training strategies proposed on balanced datasets, we compare recent adversarial training methods in Table 8. The results indicate that improving robustness on a balanced dataset is non-trivial, but these improvements cannot be expressed under an unbalanced dataset. Furthermore, we find that the simplest and the most straightforward method, PGD-AT, obtains the best results. On the other hand, methods adopting clean samples to train models, like TRADES and MART, will achieve lower clean accuracy, as the unbalanced data will harm the model's accuracy on the balanced test set.

In Figure 6, the t-SNE results prove that each class is assigned an area of a similar size in the feature space when the model is trained on balanced data. But, if the model is trained on unbalanced data,

| Method | Clean Accuracy | PGD-20 | PGD-100 | CW-100 | AA |
|---|---|---|---|---|---|
| PGD-AT | 51.28(1.34) | 29.57(0.17) | 29.47(0.15) | 29.05(0.07) | 27.71(0.15) |
| TRADES | 45.55(0.89) | 28.24(0.21) | 28.21(0.20) | 27.29(0.17) | 26.78(0.10) |
| PGD-AWP | 36.45(3.40) | 26.52(1.64) | 26.47(1.62) | 26.04(1.54) | 25.23(1.47) |
| TRADES-AWP | 41.30(0.47) | 27.04(0.08) | 27.00(0.07) | 25.65(0.08) | 25.37(0.07) |
| MART | 41.76(0.63) | 29.18(0.12) | 29.14(0.13) | 27.04(0.06) | 26.06(0.03) |

Table 8: Results on CIFAR-10-LT (UR=50) with different training strategies.

the areas for head classes expand and encroach areas that should belong to tail classes, causing the area of tail features to shrink, which represents the **unbalanced feature embedding space**. As a result, the performance and generalizability for tail classes decrease.

To alleviate the unbalance problem, we replace the cross-entropy loss in TRADES and MART with Balanced Softmax Loss (BSL). However, in our experiments, we find that BSL will make the model not converge. The reason can be that the gradient directions of BSL and KL divergence are contradicted. So, in our paper, we mainly consider enhancing the PGD-AT method to better fit the unbalanced datasets.

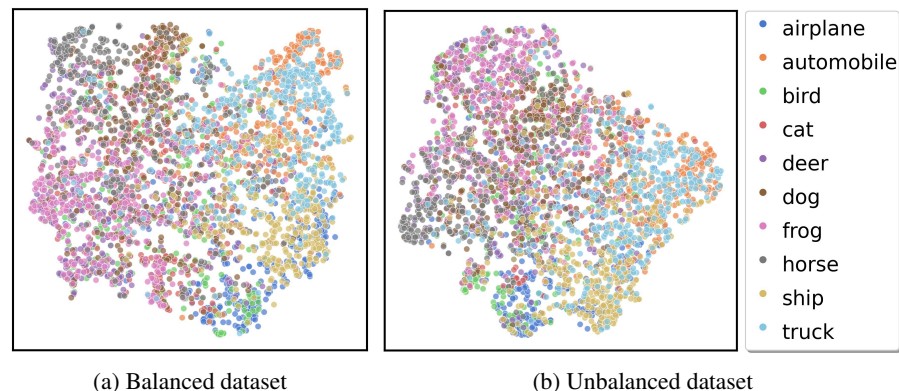

(a) Balanced dataset        (b) Unbalanced dataset

Figure 6: AE's feature t-SNE results for ResNet-18 trained with PGD-AT on balanced and unbalanced CIFAR-10.

## C  STUDYING DATA HUNGER AND DATA UNBALANCE

| UR | Method | Clean Accuracy | PGD-20 | PGD-100 | CW-100 | AA |
|---|---|---|---|---|---|---|
| 10 | PGD-AT (BS) | 77.12(0.73) | 44.73(0.20) | 44.49(0.20) | 43.81(0.26) | 41.50(0.19) |
| | PGD-AT | 75.27(0.32) | 42.66(0.20) | 42.36(0.20) | 41.18(0.21) | 38.81(0.10) |
| | REAT | 75.20(0.03) | 42.97(0.17) | 42.76(0.19) | 41.52(0.22) | 39.25(0.21) |
| 20 | PGD-AT (BS) | 75.61(0.10) | 43.37(0.15) | 43.22(0.17) | 42.12(0.12) | 40.01(0.03) |
| | PGD-AT | 72.31(0.24) | 39.79(0.31) | 39.61(0.30) | 38.42(0.06) | 36.18(0.03) |
| | REAT | 72.73(0.50) | 40.57(0.15) | 40.41(0.12) | 38.55(0.29) | 36.53(0.21) |
| 50 | PGD-AT (BS) | 72.98(0.74) | 41.14(0.26) | 40.89(0.30) | 39.92(0.49) | 37.75(0.40) |
| | PGD-AT | 66.99(0.17) | 35.23(0.45) | 35.01(0.43) | 33.17(0.37) | 31.15(0.49) |
| | REAT | 67.33(0.45) | 36.20(0.06) | 36.02(0.09) | 33.98(0.23) | 32.08(0.12) |
| 100 | PGD-AT (BS) | 72.83(0.53) | 40.25(0.35) | 40.10(0.45) | 39.29(0.19) | 37.24(0.18) |
| | PGD-AT | 62.70(0.52) | 32.91(0.17) | 32.73(0.19) | 30.45(0.15) | 28.60(0.21) |
| | REAT | 63.92(0.68) | 32.84(0.07) | 32.69(0.15) | 30.73(0.38) | 28.90(0.33) |

Table 9: Results on CIFAR-10 and CIFAR-10-LT with different URs. BSL loss is adopted for PGD-AT and REAT.

| UR | Method | Clean Accuracy | PGD-20 | PGD-100 | CW-100 | AA |
|---|---|---|---|---|---|---|
| 10 | PGD-AT (BS) | 48.32(0.50) | 20.08(0.24) | 19.95(0.25) | 18.88(0.28) | 17.44(0.21) |
|  | PGD-AT | 45.96(0.49) | 18.85(0.19) | 18.73(0.17) | 17.70(0.13) | 16.21(0.13) |
|  | REAT | 45.94(0.15) | 19.26(0.18) | 19.16(0.18) | 17.99(0.09) | 16.58(0.06) |
| 20 | PGD-AT (BS) | 45.14(0.28) | 17.95(0.19) | 17.82(0.17) | 17.15(0.19) | 15.80(0.23) |
|  | PGD-AT | 42.45(0.53) | 16.36(0.13) | 16.24(0.14) | 15.47(0.17) | 14.17(0.09) |
|  | REAT | 41.98(0.21) | 16.84(0.10) | 16.72(0.12) | 15.77(0.23) | 14.45(0.20) |
| 50 | PGD-AT (BS) | 42.86(0.37) | 16.52(0.26) | 16.38(0.24) | 15.86(0.14) | 14.54(0.05) |
|  | PGD-AT | 37.70(0.12) | 13.95(0.07) | 13.86(0.05) | 13.17(0.11) | 12.10(0.02) |
|  | REAT | 37.43(0.37) | 14.25(0.22) | 14.18(0.26) | 13.38(0.15) | 12.32(0.17) |

Table 10: Results on CIFAR-100 and CIFAR-100-LT with different URs. BSL loss is adopted for PGD-AT and REAT.

| | Method | Clean Accuracy | PGD-20 | PGD-100 | CW-100 | AA |
|---|---|---|---|---|---|---|
| ResNet | PGD-AT (BS) | 72.98(0.74) | 41.14(0.26) | 40.89(0.30) | 39.92(0.49) | 37.75(0.40) |
|  | PGD-AT | 66.99(0.17) | 35.23(0.45) | 35.01(0.43) | 33.17(0.37 | 31.15(0.49) |
|  | REAT | 67.33(0.45) | 36.20(0.06) | 36.02(0.09) | 33.98(0.23) | 32.08(0.12) |
| WRN | PGD-AT (BS) | 78.65(0.15) | 42.42(0.33) | 42.05(0.32) | 42.21(0.08) | 39.86(0.37) |
|  | PGD-AT | 72.38(0.30) | 35.93(0.10) | 35.64(0.04) | 34.93(0.14) | 32.84(0.19) |
|  | REAT | 72.58(0.31) | 36.53(0.31) | 36.35(0.32) | 35.30(0.37) | 33.37(0.37) |

Table 11: Results on CIFAR-10 and CIFAR-10-LT (UR=50) with different model structures. BSL loss is adopted for PGD-AT and REAT.

| | Method | Clean Accuracy | PGD-20 | PGD-100 | CW-100 | AA |
|---|---|---|---|---|---|---|
| ResNet | PGD-AT (BS) | 48.32(0.50) | 20.08(0.24) | 19.95(0.25) | 18.88(0.28) | 17.44(0.21) |
|  | PGD-AT | 45.96(0.49) | 18.85(0.19) | 18.73(0.17) | 17.70(0.13) | 16.21(0.13) |
|  | REAT | 45.94(0.15) | 19.26(0.18) | 19.16(0.18) | 17.99(0.09) | 16.58(0.06) |
| WRN | PGD-AT (BS) | 52.33(0.42) | 21.95(0.10) | 21.77(0.14) | 21.41(0.20) | 19.58(0.17) |
|  | PGD-AT | 50.07(0.25) | 20.79(0.39) | 20.69(0.38) | 20.17(0.27) | 18.32(0.28) |
|  | REAT | 49.99(0.18) | 20.85(0.16) | 20.71(0.20) | 20.18(0.09) | 18.35(0.17) |

Table 12: Results on CIFAR-100 and CIFAR-100-LT (UR=10) with different model structures. BSL loss is adopted for PGD-AT and REAT.

In this part, we further examine the effects of the data hunger and the data unbalance on the model robustness, which is explored to construct an experimental upper bound on the robustness of the long-tailed adversarial training methods. To be specific, the data hunger raises from the insufficient data from the body classes and tail classes, which is one of the impacts of the long-tailed datasets. And another one is the data unbalance. To exclusively study the data hunger in a balanced dataset, for a given unbalanced ratio, we sample the same number of samples as the long-tail dataset but form them into balanced small (BS) datasets. We then train models on this dataset with PGD-AT to learn an experimental upper bound, which is represented as "PGD-AT (BS)" in our experiments. When we train models with PGD-AT (BS) the loss function used to optimize models is Cross-Entropy loss. On the other hand, when we train models on unbalanced datasets, the basic loss function used to optimize models is BSL.

Comparing the results of models trained under balanced datasets and unbalanced datasets in Tables 9–12, it is clear that the models train on unbalanced datasets suffer from a bigger reduction when the number of training samples decreases, which means that the data unbalance harms the model's robustness in a larger degree than the data hunger. Training models on unbalanced data is more challenging than training models on small but balanced data under adversarial scenarios for different model structures and datasets. It is reasonable, because in the long-tailed datasets, there are fewer data in the tail classes, making the model unable to learn much information for such classes. Furthermore, compared with training a model on CIFAR-10, when training a model on a more complex dataset,

such as CIFAR-100, the performance decrease is less than expected, which will be studied in our future work. On the other hand, the experimental results of PGD-AT (BS) can be seen as upper bounds for the models trained on same-size unbalanced datasets.

To theoretically study the difficulties of adversarially training a model on unbalanced datasets, we first consider the robust risk $\mathcal{R}_{\text{rob}}(\theta)$, where $\theta$ stands for the parameters of a function $f$, which can be a deep learning model. Let $\mathcal{X}_i \subset \mathbb{R}^d$ be the input space for class $i$, and $\mathcal{Y} = \{1, \dots, C\}$ be the class set. We consider a following dataset $\mathcal{D} = \{(x, i) | x \in \mathcal{X}_i, i \in \mathcal{Y}\}$, that for class $i$, we only have $n_i$ data in it. Then, we define that $p_\theta(\cdot|x) = \text{softmax}(f_\theta(x)) \in \mathbb{R}^C$ and $F_\theta(x) = \text{argmax}[f_\theta(x)] \in \mathcal{Y}$. Given a perturbed set $\mathcal{B}_p(x_i, \epsilon) = \{x_i' \in \mathcal{X}_i | \|x_i - x_i'\|_p \leq \epsilon\}$, we have $\mathcal{R}_{\text{rob}}(\theta) = \mathbb{E}_\mathcal{D}[\max_{x_i' \in \mathcal{B}_p(x_i, \epsilon)} \mathbb{1}\{F_\theta(x_i') \neq i\}]$. Assume that for each class $i$, its $p_\theta(\cdot|x_i)$ is independent to other class $j \neq i$. We have $\mathcal{R}_{\text{rob}}(\theta) = \sum_{i=1}^C \frac{1}{n_i} \sum_{j=1}^{n_i} \max_{x' \in \mathcal{B}_p(x_i^j, \epsilon)} \mathbb{1}\{F_\theta(x') \neq i\}p(i)$. Let $\mathcal{R}_{\text{rob}}^i(\theta) = \sum_{j=1}^{n_i} \max_{x' \in \mathcal{B}_p(x_i^j, \epsilon)} \mathbb{1}\{F_\theta(x') \neq i\}p(i)$, which can be further decomposed of two terms, i.e., the natural risk and the boundary risk (Zhang et al., 2019), i.e., $\mathcal{R}_{\text{rob}}^i(\theta) = \mathcal{R}_{\text{nat}}^i(\theta) + \mathcal{R}_{\text{bdy}}^i(\theta) = \sum_{j=1}^{n_i} \mathbb{1}\{F_\theta(x_i^j) \neq i\}p(i) + \sum_{j=1}^{n_i} \mathbb{1}\{\exists x' \in \mathcal{B}_p(x_i^j, \epsilon), F_\theta(x') \neq i\}p(i)$. We notice that for an unbalanced dataset, $p(i) = \frac{n_i}{\sum_{i=1}^C n_i}$. Therefore, we have the following lower bound for the robust risk $\mathcal{R}_{\text{rob}}(\theta)$.

**Theorem 1** *Given a function $f_\theta$ and dataset $\mathcal{D}$, which we defined above, let*

$$g(x) \in \text{argmax}_{x' \in \mathcal{B}_p(x, \epsilon)} \mathbb{1}\{F_\theta(x) \neq F_\theta(x')\}$$

*Assume that a larger $n_i$ will decrease the $\mathcal{R}_{\text{rob}}^i(\theta)$, and vice versa. Then, we have*

$$\mathcal{R}_{\text{rob}}(\theta) \geq \sum_{i=1}^C \frac{1}{C} \left( \sum_{j=1}^{n_i} \mathbb{1}\{F_\theta(x_i^j) \neq i\} + \sum_{j=1}^{n_i} \mathbb{1}\{F_\theta(g(x_i^j)) \neq F_\theta(x_i^j)\} \right)$$

**Proof 1** *We have*

$$\mathcal{R}_{\text{bdy}}^i(\theta) = \sum_{j=1}^{n_i} \mathbb{1}\{\exists x' \in \mathcal{B}_p(x_i^j, \epsilon), F_\theta(x') \neq i\}p(i)$$

$$\geq \sum_{j=1}^{n_i} (\mathbb{1}\{F_\theta(g(x_i^j)) \neq F_\theta(x_i^j), F_\theta(x_i^j) = i\} + \mathbb{1}\{F_\theta(g(x_i^j)) \neq F_\theta(x_i^j), F_\theta(x_i^j) \neq i\})p(i)$$

$$= \sum_{j=1}^{n_i} (\mathbb{1}\{F_\theta(g(x_i^j)) \neq F_\theta(x_i^j)\}\mathbb{1}\{F_\theta(x_i^j) = i\} + \mathbb{1}\{F_\theta(g(x_i^j)) \neq F_\theta(x_i^j)\}\mathbb{1}\{F_\theta(x_i^j) \neq i\})p(i)$$

*Let $\mathcal{A} = \mathbb{1}\{F_\theta(x_i^j) \neq i\}$ and $\mathcal{B} = \mathbb{1}\{F_\theta(g(x_i^j)) \neq F_\theta(x_i^j)\}$, we have*

$$\mathcal{R}_{\text{rob}}(\theta) \geq \sum_{i=1}^C p(i) \left( \sum_{j=1}^{n_i} \mathcal{A} + \mathcal{B}(1 - \mathcal{A}) + \mathcal{B}\mathcal{A} \right)$$

$$= \sum_{i=1}^C p(i) \left( \sum_{j=1}^{n_i} \mathcal{A} + \mathcal{B} \right)$$

*Based on the assumption and Chebyshev inequality, we have*

$$\mathcal{R}_{\text{rob}}(\theta) \geq \sum_{i=1}^C \frac{1}{C} \left( \sum_{j=1}^{n_i} \mathcal{A} + \mathcal{B} \right)$$

$$= \sum_{i=1}^C \frac{1}{C} \left( \sum_{j=1}^{n_i} \mathbb{1}\{F_\theta(x_i^j) \neq i\} + \sum_{j=1}^{n_i} \mathbb{1}\{F_\theta(g(x_i^j)) \neq F_\theta(x_i^j)\} \right)$$

In Theorem 1, $\mathbb{1}\{F_\theta(x_i^j) \neq i\}$ is the error for clean data $x_i^j$, and $\mathbb{1}\{F_\theta(g(x_i^j)) \neq F_\theta(x_i^j)\}$ is a general robust error, which means that the prediction of the perturbed data disagrees with the prediction of the clean data. Based on Theorem 1, we have the conclusion that the lower bound of the robust risk on an unbalanced dataset is the robust risk of the corresponding balanced one, which has the same number of data as the unbalanced one. Therefore, we provide a theoretical robustness lower bound for the unbalanced dataset.

## D    RESULTS UNDER $l_2$-NORM ATTACKS

| UR | Method | Clean Accuracy | PGD-20 | PGD-100 | CW |
|---|---|---|---|---|---|
| 10 | PGD-AT | 75.27(0.32) | 30.92(0.47) | 28.82(0.63) | 70.76(0.20) |
| | RoBal | 75.33(0.39) | 33.15/28.26(0.62) | 31.43/26.18(0.80) | 71.36(0.39) |
| | REAT | 75.20(0.03) | 30.82(0.60) | 28.71(0.74) | 71.02(0.18) |
| 20 | PGD-AT | 72.31(0.24) | 29.23(0.50) | 27.24(0.55) | 67.87(0.26) |
| | RoBal | 71.92(0.62) | 31.79/27.13(0.37) | 30.34/25.38(0.44) | 67.95(0.58) |
| | REAT | 72.73(0.50) | 29.27(0.58) | 27.44(0.59) | 68.13(0.43) |
| 50 | PGD-AT | 66.99(0.17) | 26.75(0.31) | 25.20(0.31) | 62.54(0.14) |
| | RoBal | 66.08(0.69) | 29.22/24.17(0.68) | 27.97/22.83(0.60) | 62.03(0.65) |
| | REAT | 67.33(0.45) | 27.45(0.30) | 25.91(0.40) | 63.08(0.23) |
| 100 | PGD-AT | 62.70(0.52) | 24.91(0.46) | 23.49(0.42) | 58.06(0.40) |
| | RoBal | 60.11(0.62) | 27.77/23.33(0.46) | 26.48/21.91(0.43) | 56.34(0.28) |
| | REAT | 63.92(0.68) | 24.63(0.21) | 23.24(0.21) | 59.17(0.49) |

Table 13: Results on CIFAR-10-LT with different URs under $l_2$-norm attacks. Red numbers represent the results under our adaptive attack. BSL loss is adopted for PGD-AT and REAT.

In Table 13, we show the results of models under $l_2$-norm attacks. For the PGD attacks, the max perturbation size is $\epsilon = 1.0$, and the step length is $\alpha = 0.2$. We consider the 20-step attack, PGD-20, and the 100-step attack, PGD-100. For the C&W attack, we follow its official implementation. The results confirm that our REAT can improve the model's robustness under different threat models. On the other hand, the gradient obfuscation is more serious under $l_2$-norm attacks, so our adaptive attacks achieve better results.

## E    COMPARING WITH ROBAL

| Method | | Clean Accuracy | PGD-20 | PGD-100 | CW-100 | AA |
|---|---|---|---|---|---|---|
| ResNet | RoBal | 66.08(0.69) | 38.46(0.18) Adaptive: 33.54 | 38.44(0.11) Adaptive: 33.20 | 33.90(1.72) | 31.14(0.44) |
| | REAT | 67.33(0.45) | 36.20(0.06) | 36.02(0.09) | 33.98(0.23) | 32.08(0.12) |
| WRN | RoBal | 69.33(0.11) | 39.97(0.30) Adaptive: 34.58 | 39.98(0.32) Adaptive: 34.26 | 34.83(0.21) | 33.09(0.41) |
| | REAT | 72.58(0.31) | 36.53(0.31) | 36.35(0.32) | 35.30(0.37) | 33.37(0.37) |

Table 14: Results on CIFAR-10-LT (UR=50) with different model structures. Red numbers represent the results under our adaptive attack. BSL loss is adopted for REAT.

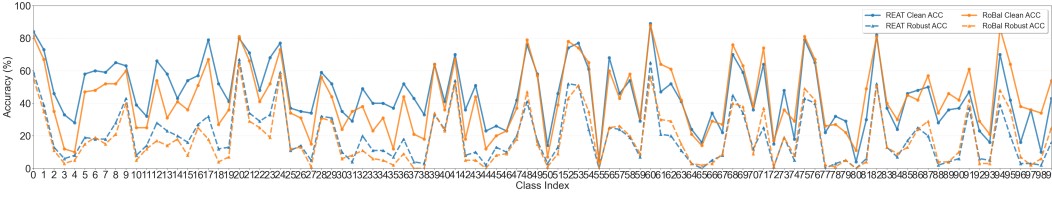

Figure 8: Accuracy of Each Class on CIFAR-100-LT (UR=10).

| Method | | Clean Accuracy | PGD-20 | PGD-100 | CW-100 | AA |
|---|---|---|---|---|---|---|
| ResNet | RoBal | 43.47(0.31) | 20.55(0.20)
Adaptive: 18.49 | 20.49(0.20)
Adaptive: 18.31 | 18.12(0.23) | 16.86(0.10) |
| | REAT | 45.94(0.15) | 19.26(0.18) | 19.16(0.18) | 17.99(0.09) | 16.58(0.06) |
| WRN | RoBal | 48.84(0.24) | 21.45(0.18)
Adaptive: 19.29 | 21.44(0.21)
Adaptive: 19.19 | 19.71(0.09) | 18.21(0.02) |
| | REAT | 49.99(0.18) | 20.85(0.16) | 20.71(0.20) | 20.18(0.09) | 18.35(0.17) |

Table 15: Results on CIFAR-100-LT (UR=10) with different model structures. Red numbers represent the results under our adaptive attack. BSL loss is adopted for REAT.

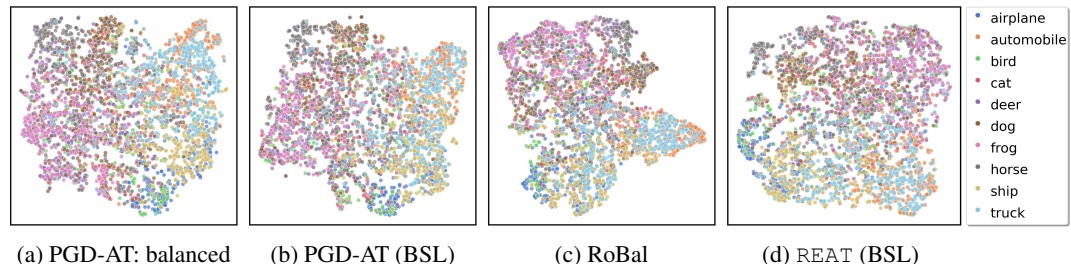

(a) PGD-AT: balanced     (b) PGD-AT (BSL)     (c) RoBal     (d) REAT (BSL)

Figure 9: AE's feature map results with different strategies. (a) is trained with the balanced dataset (CIFAR-10) while the rest three are trained with the unbalanced dataset (CIFAR-10-LT, UR=50).

**Varying Datasets**. Similar to our main paper, we illustrate the robust accuracy under the different numbers of PGD attack steps of ResNet-18 on CIFAR-100-LT (UR=10) in Figure 7 in this section. The results prove that our REAT outperforms RoBal under all attack budgets. In Figure 8, we plot the accuracy of ResNet-18 on CIFAR-100-LT (UR=10) under the PGD-20 attack for each class. The results indicate that our REAT can achieve higher clean accuracy and robust accuracy on "body" classes, which is consistent with the conclusion in the main paper.

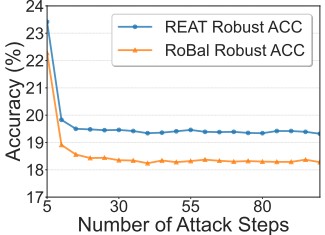

Figure 7: Accuracy of Attack Steps on CIFAR-100-LT (UR=10).

**Varying Model Structures**. To show the superiority of REAT on different model structures, we compare the results of RoBal and REAT on ResNet-18 and WideResNet-28-10, respectively. The results in Tables 14 and 15 prove that models trained with REAT lead models trained with RoBal on both clean accuracy and robustness, which means REAT is a better training strategy for different model structures.

**Interpretation**. We choose the configurations of CIFAR-10-LT (UR=50) and ResNet-18. Then, we plot the feature embedding space with the t-SNE tool for models trained with different strategies in Figure 9. We first generate AEs with the PGD-20 attack on the test set and use t-SNE to plot the feature distribution for AEs. ResNet-18 is adopted as the model architecture. Figure 9a is the feature result for PGD-AT over the balanced dataset CIFAR-10. We observe that samples from different classes are not quite overlapped with each other in the feature space, making them easier to be classified. In contrast, Figures 9b and 9c show the results for PGD-AT (BSL loss) and RoBal over the unbalanced dataset CIFAR-10. We observe that there are more samples from different classes entangled together in their feature embeddings, which can harm the model's robustness. Figure 9d shows the results of our REAT under the same unbalanced setting. We can see the feature space is more similar to the one obtained from the balanced dataset (Figure 9a). This explains the effectiveness of REAT in enhancing the model robustness and clean accuracy from the feature perspective.

# F   AE PREDICTION DISTRIBUTION

In Figure 10, the distributions of the model's predictions for generated AEs in different epochs are illustrated. From the plot, we can obtain the same conclusion as the one in our main paper that the RoBal will make the model pay more attention to head and tail classes and ignore the body classes.

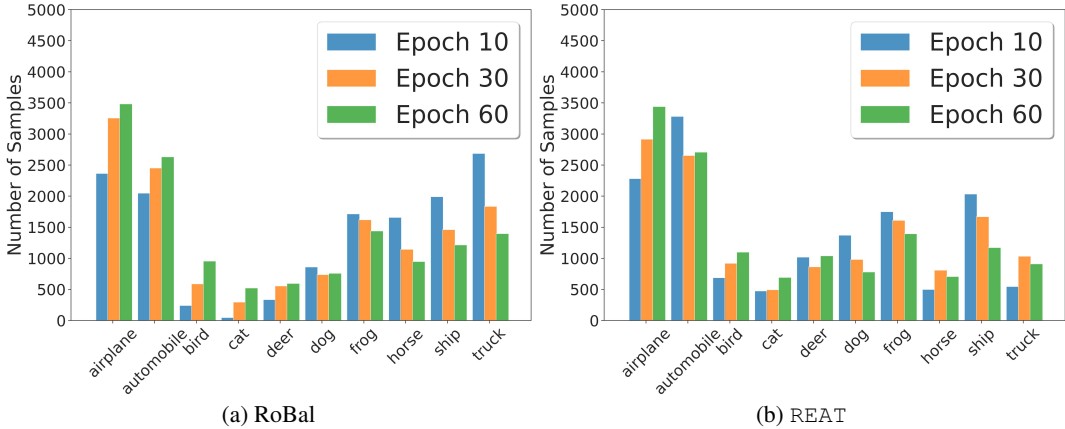

Figure 10: The distribution of model predictions for AEs in Epoch {10, 30, 60} on CIFAR-10-LT (UR=50).

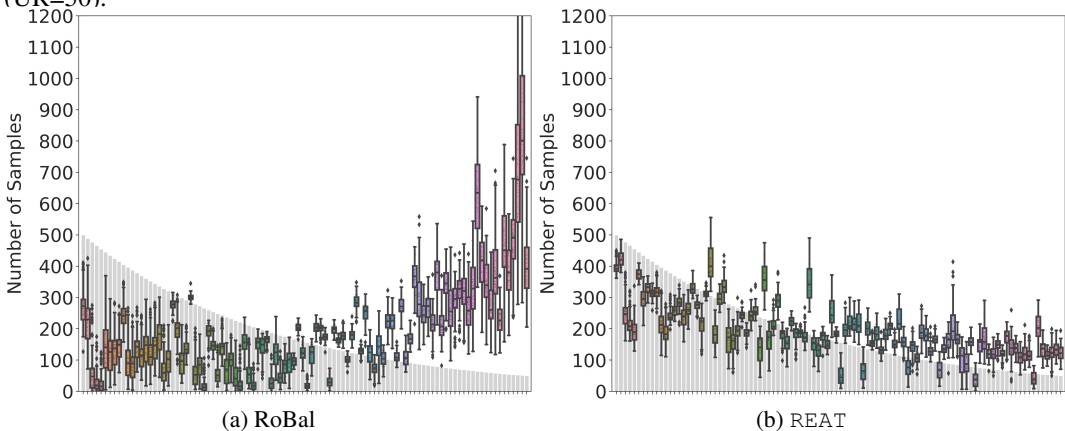

Figure 11: The distribution of model predictions for AEs during the training process on CIFAR-100-LT (UR=10). Clean label distributions are shown by gray bars.

On the contrary, our REAT will help the model value the body and tail classes equally, which can further improve the model's robustness.

In Figure 11, we compare the AE distribution on CIFAR-100-LT (UR=10), when training models with RoBal and REAT, respectively. The results prove that RoBal will cause unbalanced AE distribution when the number of classes increases. There are more samples predicted as tail classes by the model. However, our REAT can keep the balanced AE distribution and obtain better results.

## G  TRAINING COST OVERHEAD OF REAT

We compare the training time overhead of REAT compared with PGD-AT (BSL is adopted) method and RoBal on one single V100 GPU card. The results are shown in Table 16. When we train a ResNet-18 on CIFAR-10-LT (UR=50), the training time overhead for one epoch is about 8 seconds. When we train a ResNet-18 on CIFAR-10-LT (UR=100), the training time overhead for one epoch is about 5 seconds. So, our REAT is efficient on long-tailed datasets and does not increase too much training time.

| Dataset | UR | Time Cost (Secs) | | |
|---|---|---|---|---|
| | | PGD-AT | RoBal | REAT |
| CIFAR-10-LT | 10 | 68 | 73 | 96 |
| | 20 | 57 | 61 | 74 |
| | 50 | 46 | 50 | 54 |
| | 100 | 42 | 45 | 47 |
| CIFAR-100-LT | 10 | 65 | 70 | 99 |
| | 20 | 53 | 58 | 75 |
| | 50 | 42 | 45 | 52 |

Table 16: Time cost (seconds) for one training epoch on CIFAR-10-LT and CIFAR-100-LT. The model is ResNet-18. BSL loss is adopted for PGD-AT and REAT.

