# OpenReview forum: "Alleviating the Effect of Data Imbalance on Adversarial Training"
_ICLR.cc/2024/Conference — ICLR 2024 Conference Withdrawn Submission_

### Official Review · Reviewer_prkd · 2023-10-24

**Soundness:** 3 good
**Presentation:** 3 good
**Contribution:** 2 fair
**Rating:** 5
**Confidence:** 2

**Summary:**

This paper tackles the domain of adversarial training for imbalanced dataset. For this purpose they propose the REAT framework. REAT has two main contributions. First, it tackles the imbalance of AE generation by employing a re-weighting scheme using the class effective number. This helps by increasing the number adversarial examples belonging to the tail classes. Next, they introduce a regularization term called TAIL to re-balance the feature distribution across classes. Finally, the authors demonstrate the efficacy of REAT through multiple experiments.

**Strengths:**

1. The domain this paper is tackling is an important one. The motivation of the two main contributions of this paper (re-weighting and TAIL) are clear. In particular, the inclusion of effective number to handle AE generation is unique and well formulated.
2. The empirical studies (in particular the ablation studies) display the efficacy of proposed method very well.

**Weaknesses:**

1. The authors provide experimental results that show the efficacy of the proposed method. However, the improvements are not consistent across the board. For example, the clean accuracy of REAT always seems to be lower than the baselines at UR = 10. Similarly the adversarial accuracy of REAT is not always better and in many cases the improvements are marginal. The experiments are also limited to the CIFAR-10-LT and CIFAR-100-LT dataset. Experimenting with further datasets would have been helpful.
2. The proposed REAT method is intuitive to understand but it would have been better if there were some theoretical justification behind performance improvement. For example, the authors assume that predicted label distribution will not change much between two successive epochs but no reasoning is provided behind this assumption.

3. There are multiple typos and grammatical mistakes in the manuscript.

**Questions:**

1. I want to know the authors thoughts about why REAT's clean accuracy is usually lower at UR = 10
2. The authors stated that REAT can handle "body" classes better. I might have missed this but I didn't see a clear justification regarding why REAT would do this better than RoBal

---

### Official Review · Reviewer_1nCF · 2023-10-30

**Soundness:** 3 good
**Presentation:** 3 good
**Contribution:** 3 good
**Rating:** 6
**Confidence:** 3

**Summary:**

This paper focuses on adversarial training with imbalanced datasets. It addresses two main motivations: (1) The prediction distribution of adversarial example generation during adversarial training is imbalanced; and (2) the feature space of the tail classes is smaller than that of the head classes. The paper proposes corresponding solutions to improve adversarial training in scenarios with data imbalance.

**Strengths:**

1. This paper focuses on an important but somewhat overlooked problem in adversarial training (AT): the imbalanced dataset setting.
2. This paper is clearly written and well-organized. The first two sections detail the introduction of related work and the motivation of the proposed method.
3. The motivations of the proposed method, observations on the prediction distribution of adversarial examples generation during AT, and feature embeddings are clear and supported by empirical validation and theoretical analysis.
4. The experimental settings are clear and diverse.

**Weaknesses:**

1. Adversarial training is known to result in a robust fairness issue. This means that even in a balanced dataset, the robustness of different classes varies significantly, which can be seen as an inherent data imbalance effect. Therefore, it would be beneficial to compare the proposed methods [2, 3] that address this issue (or at least mention them in the related work).
2. The improvement of the proposed method appears to be limited compared to RoBal [1]. While this is not a major concern, it would be interesting to explore if combining RoBal and REAT can further enhance robustness.
3. The experiments only focus on CIFAR-10 and CIFAR-100 datasets. It is recommended to include larger datasets like TinyImagenet to further validate the effectiveness of REAT.

[1] Adversarial Robustness Under Long-Tailed Distribution. CVPR

[2] To be robust or to be fair: Towards fairness in adversarial training. ICML

[3] CFA: Class-wise calibrated fair adversarial training. CVPR

**Questions:**

1. Can REAT also mitigate the robust fairness issue in adversarial training on **balanced** datasets?
2. Can REAT be combined with Robal to further mitigate the effect of data imbalance on adversarial training?
3. Please provide more details on the application of Re-Balancing Loss (*RBL*) in equations (1) and (2). How is this loss function used to generate adversarial examples?

---

### Official Review · Reviewer_yNqS · 2023-10-30

**Soundness:** 3 good
**Presentation:** 4 excellent
**Contribution:** 2 fair
**Rating:** 3
**Confidence:** 4

**Summary:**

This paper studies the adversarial training method under imbalanced dataset. In order to improve the model performance, the paper proposes two strategies: (1) reweighting and (2)  tail feature alignment

**Strengths:**

The paper is clearly written and the proposed methods seems reasonable.

**Weaknesses:**

I have the following concerns:
1. The performance improvement is very marginal compared to the original AT.
2. There is no evidence provided to show why the tail feature alignment is necessary. For example, whether this will cause performance compromise is not clear.
3. Similar studies are investigated previously, i.e., the paper [1].

[1] Imbalanced adversarial training with reweighting, Wang et al, 2022.

**Questions:**

Plz see the question above.

---

### Official Review · Reviewer_h8eU · 2023-11-04

**Soundness:** 3 good
**Presentation:** 3 good
**Contribution:** 2 fair
**Rating:** 5
**Confidence:** 3

**Summary:**

The authors study the long-tailed distribution problem in the adversarial training scenario and show the drawbacks of unbalanced AE and feature space. Based on this insight, the authors propose a two-stage framework to improve the performance with unbalanced datasets.

**Strengths:**

The authors show that the head class dominates the AE label and feature embedding space, leading to the underfitting of the tail class.

The author proposed RBL and TAIL corresponding to the maximization and minimization processes respectively to jointly solve this problem.

This paper is well-written and easy to understand, and the experimental results are extensive.

**Weaknesses:**

Fig. 3 shows the effectiveness of RBL is not signifient, could introducing a hyperparameter further accentuate the weight disparities to enhance robustness?

The TAIL only improve the TC’s feature space, could TAIL be tuned to align the feature space for all classes, e.g. class-dependent maximizing the KL divergence between each class and the first head class?

The performance improvement of the REAT framework is relatively weak, using 25\% additional computing overhead to improve performance by 1\% compared to the baseline.

**Questions:**

Shown in weaknesses.